# The Fate of Entanglement

Gilles Parez[1,2] and William Witczak-Krempa[1,2,3]

[1]*Département de Physique, Université de Montréal, Montréal, QC H3C 3J7, Canada*

[2]*Centre de Recherches Mathématiques, Université de Montréal, Montréal, QC H3C 3J7, Canada*

[3]*Institut Courtois, Université de Montréal, Montréal, QC H2V 0B3, Canada*

Quantum entanglement manifests itself in non-local correlations between the constituents of a system. In its simplest realization, a measurement on one subsystem is affected by a prior measurement on its partner, irrespective of their separation. For multiple parties, purely collective types of entanglement exist but their detection, even theoretically, remains an outstanding open question. Here, we argue that all forms of multipartite entanglement entirely disappear during the typical evolution of a physical state as it heats up, evolves in time in a large family of dynamical protocols, or as its parts become separated. We focus on the generic case where the system interacts with an environment. These results mainly follow from the geometry of the entanglement-free continent in the space of physical states, and hold in great generality. We illustrate these phenomena with a frustrated molecular quantum magnet in and out of equilibrium, and a quantum spin chain. In contrast, if the particles are fermions, such as electrons, another notion of entanglement exists that protects bipartite quantum correlations. However, genuinely collective fermionic entanglement disappears during typical evolution, thus sharing the same fate as in bosonic systems. These findings provide fundamental knowledge about the structure of entanglement in quantum matter and architectures, paving the way for its manipulation.

## I. INTRODUCTION

Entanglement is a fundamental property of quantum mechanics which can manifest itself in non-local quantum correlation between two or more systems. In particular, it makes it possible for a measurement on a subset of the whole system to affect subsequent measurements on the other parties. The effect of the initial measurement is instantaneous, even if the subsystems are distant. Entanglement not only constitutes a fundamental property of nature, but it is also a resource to perform tasks that would prove impossible without it such as teleportation [1, 2], or more broadly quantum computation [3–5]. This has driven the community to devise methods to detect and quantify entanglement [6–8]. Unfortunately, it it not known how to determine with certainty whether a general system is entangled, except in very simple situations such as with 2 qubits. One can better grasp the complexity of the task by observing that entanglement can

exist between more than two parties. In fact, some systems possess 3-party entanglement but no 2-party entanglement of any sort, such as certain mixed states constructed from unextendible product bases [9].

In this work, we identify conditions for the existence of multipartite entanglement in quantum many-body systems such as spin models, the fermion Hubbard model, local quantum circuits, etc. We allow the state to be interacting with an environment: this can take the form of a thermal bath, or the neighboring spins/fermions when one considers a subregion. For instance, we answer the question: at what separations can multipartite entanglement of a given kind exist in a quantum many-body system/material? We begin by explaining important properties about the space of physical states, and how these determine the fate of entanglement under the evolution of a system with temperature, time or separation. We then illustrate these results with a simple yet generic model: the frustrated anti-ferromagnetic Ising model on an icosahedral molecule. Finally, we discuss the fate of entanglement for fermionic systems, where the fermion parity superselection rule modifies the geometry of the space of states and the structure of entanglement.

## II. SEPARABLE SET

We investigate multipartite entanglement of states with $m$ subsystems, as illustrated in Fig. 1a. We shall argue that in numerous physical situations, the end point of the evolution typically corresponds to an un-entangled state, which is called separable. The simplest separable state for a system of $m$ parties is a product,

$$\rho_{\mathrm{prod}} = \rho_1 \otimes \rho_2 \otimes \cdots \otimes \rho_m, \tag{1}$$

where $\rho_j$ is a physical density matrix for subsystem $j$. The set of separable sates is convex [6], namely states composed of a mixture of product states, $\rho_{\mathrm{sep}} = \sum_k p_k \rho_{\mathrm{prod}}^{(k)}$ with $p_k \geqslant 0$ and $\sum_k p_k = 1$, are also un-entangled. In the space of all physical states, separable ones thus form a "continent" surrounded by an "ocean" of entangled states, see Fig. 1b. We stress however that this representation does not account for the intricate geometry of the boundary, but it illustrates the relevant ingredients for our discussion, namely the fact that the set of separable states is convex and has a non-vanishing volume. In our schematic representation, highly-entangled states live in the deep-blue regions, whereas in the center of the separable continent lies the maximally-mixed state, i.e., the identity. In the following, we focus on finite and discrete subsystems, and thus on finite-dimensional local density matrices. In the thermodynamic limit, the order of limits is chosen

such that the total system remains incommensurably larger than the finite subsystems, which can in turn be large.

We describe the system's state by $\rho(s)$, where $s$ parametrizes the evolution; the final state is $\rho_f \equiv \rho(s_f)$, and we typically (but not always) consider $s_f = \infty$. We shall argue that in numerous situations of physical relevance the final state is separable. Moreover, $\rho_f$ typically lies in the interior of the separable continent, not on its frontier. Under these conditions, we thus arrive at our main conclusion: in numerous physically relevant situations, the system irreversibly looses all forms of multipartite entanglement beyond some stage in the evolution, as shown in Fig. 1b. Moreover, before reaching land, the system navigates shallow waters, leading to a rapid decimation of entanglement. During this stage one has an effectively separable state. This constitutes a point of significant importance in real-world applications since determining when a state reaches a very low degree of entanglement is easier than showing exact separability. Indeed, numerical estimations of the degree of entanglement typically involve optimization procedures which correctly yield small results for low-entangled states, but do not necessarily perfectly detect exact separability. This phenomenon notably occurs for the geometric entanglement, discussed later in the text. Nevertheless, we can exploit a mathematical result that tells us when a state is in the interior of the separable continent [10–13]. In the simplest version, the theorem states that a product state (1) lies in the interior if it has full rank, i.e., none of its eigenvalues vanishes. This is coherent with what is known for pure product states: these have minimal rank (a single non-zero eigenvalue), and indeed live on the frontier of the continent as arbitrarily weak perturbations can make them entangled. Furthermore, the theorem yields the radius of a ball in the space of states that lies entirely on the separable continent; this is represented by the dashed line in Fig. 1b. The radius of the ball is proportional to the smallest eigenvalue of the state [13], $R_m = 2^{1-m/2}\lambda_{\min}$. Here, we use the standard notion of distance between two quantum states given by the Frobenius norm: $d(\rho, \rho') = \sqrt{\text{Tr}(\rho - \rho')^2}$. A more general version of the result holds for a mixture of product states with a least one having full rank [13]. This implies that the boundary of the separable continent does not only contain pure product states, but also higher-rank and full-rank separable states whose product-state components are all rank-deficient.

## A.  Biseparable set and genuine multipartite entanglement

In multiparty states, there is *genuine multipartite entanglement* (GME) if the state is not biseparable, i.e., if it *cannot* be written as a convex combination of product states over bipartitions

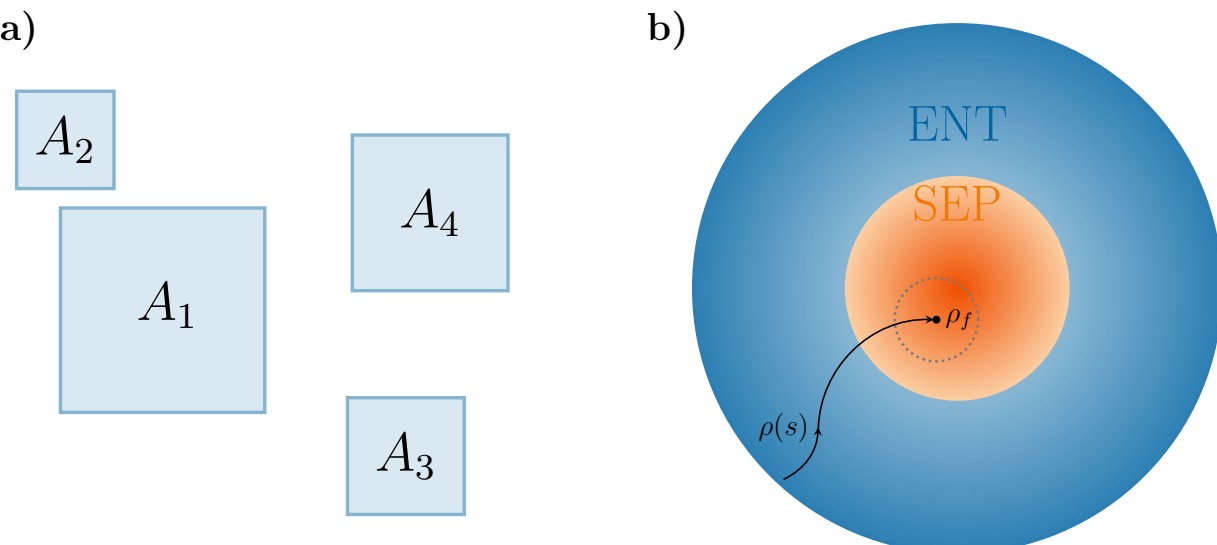

FIG. 1. **Evolution of multipartite entanglement. a)** We consider a general state of $m$ subsystems, here illustrated for $m = 4$, which can be in contact with an environment. **b)** The state is described by a density matrix $\rho(s)$ that evolves according to a parameter $s$ such as temperature, time or separation. The blue region represents the "sea" of entangled states, where deep-blue regions are more entangled than light-blue ones. The orange disk is the separable continent.

of the $m$ parties,

$$\rho_{\text{bisep}} = \sum_k p_k \, \rho_{\mathcal{I}_k} \otimes \rho_{\overline{\mathcal{I}}_k}, \tag{2}$$

where $\mathcal{I}_k \cup \overline{\mathcal{I}}_k$ are bipartitions labelled by $k$, $p_k \geqslant 0$, and $\sum_k p_k = 1$. As an example, for $m = 3$ parties, the possible bipartitions are $(1, 23)$, $(12, 3)$ and $(13, 2)$. Biseparable states contain at most $(m-1)$-party entanglement, which is not genuinely multipartite. Similarly to separable states, biseparable ones form a convex set which includes the separable continent. Using the same arguments as above, we conclude that during typical generalized evolutions $\rho(s)$, the state irreversibly loses GME at a stage prior to the sudden death of entanglement, implying that non-GME and bipartite entanglement are more robust than GME.

### B. Temperature

As a warmup case that will be a reference point when we shall discuss thermalisation, we take the parameter $s$ to represent the temperature $T$, as would pertain to a Gibbs thermal state associated with a Hamiltonian $H$, $\rho(T) = \exp(-H/T)/\mathcal{Z}$, or a reduced density matrix thereof $\rho_A(T) = \text{Tr}_B \, \rho(T)$. Naturally, a very large temperature destroys entanglement, yielding

a maximally uncorrelated state. The infinite-temperature end point is a full-rank product state, $\rho_f = D^{-1}\mathbb{I}$, where $\mathbb{I}$ represents the identity matrix in the $D$-dimensional space of the system. This state is located at the center of the separable continent. Hence, there exists a threshold temperature $T_m$ at which one can rigorously conclude that all forms of entanglement between the $m$ subsystems disappear, in agreement with previous results for finite-dimensional systems [14]. At the threshold, the state at the boundary of the separable continent will generically have full-rank albeit being in a mixture of only rank-deficient product states. The minimal eigenvalue of $\rho_f$ being $1/D$, one can readily obtain an upper bound for the temperature above which entanglement is lost. However, the example below will show that physical many-body systems tend to loose entanglement much faster than this upper bound. Sudden death of bipartite entanglement at finite temperature has been observed in different physical systems [15–19]. Later, we will contrast the above behavior with the thermal fate of entanglement in fermion systems.

## C.   Time

In a dynamical situation, the parameter $s$ is the time $t$, and we take the final state to be the state at some time $t_f$. In dynamical protocols where the system reaches a final state $\rho_f$ which lies inside the separable continent, one can conclude that there is a sudden-death time $t_m^*$ at which all $m$-party entanglement is lost until $t_f$. As the structure of the final state depends on the type of dynamical evolution under study, let us describe an important case in more detail: a global quantum quench. One prepares a closed system to be in an eigenstate of a given Hamiltonian $H_I$, and at some time the Hamiltonian is abruptly changed to a different one, $H$, resulting in non-trivial time evolution. We then study the state of a subregion $A$ of the entire system: the $m$-party state $\rho(t)$ is obtained by partially tracing over the unobserved part, $B$. When $A$ is small compared to its complement $B$, the state is expected to effectively thermalize at large times [20–22]. As discussed above, temperature tends to destroy entanglement, which means that in numerous quench protocols the subregion final state $\rho(t_f)$ will be separable. Although not all quenches will lead to a sudden death of multipartite entanglement, one can ask about the typicality of such a fate. For instance, if the initial condition has an energy expectation value close to the groundstate of $H$, instead of thermalisation, oscillatory behavior with persistent entanglement can occur. However, only a minority of eigenstates of a local Hamiltonian $H$ have an energy near the minimum (or maximum) of the spectrum: the bulk of the levels lies near the middle of the spectrum. As such, a typical initial condition will inject enough energy so that the evolution will scramble the quantum information encoded within $A$. In

many cases one can be more precise: the state of $A$ at large times is accurately approximated by an effective thermal Gibbs state, $\text{Tr}_B(\exp(-H/T_{\text{eff}}))/\mathcal{Z}$. When the effective temperature $T_{\text{eff}}$ is not small (which is typical), an entanglement sudden death for the subregion follows. Below we shall rigorously verify the dynamical sudden death with numerous examples. Another important class of dynamical protocol are the local quenches, where the Hamiltonian is modified locally, for instance by connecting two different disjoint groundstates at $t = 0$. In this case, the difference between the energy expectation value and the post-quench groundstate energy is not extensive, thus corresponding to a lower effective temperature in the steady state. It is an important question to understand whether this feature allows local quenches to generically escape the dynamical sudden death of entanglement, which we leave for future research.

To complement the discussion of late-time dynamics, we can argue in full generality that the multiparty entanglement dynamics after a global quench from a pure product state typically follows a rise-and-fall behavior. For early times, we consider the evolution of the system with parameter $s = t^{-1}$. The "final" state of this evolution is the initial state of the quench, namely a pure product state with rank one, i.e., not full rank. As we previously discussed, such states lie on the boundary, or the shore, of the separable continent. Therefore, there is no entanglement sudden death in this evolution. Looking back at the quench protocol with time as the parameter, this implies that entanglement is generated at $t = 0^+$ after the quench. As discussed above, in numerous situations, the state typically lands on the continent at late times, or in very shallow waters at a finite time, yielding an entanglement sudden death or strong suppression. These early- and late-time behaviors result in the rise-and-fall dynamics of entanglement, which have been observed for simple entanglement measures in various systems [23–29].

### D.   Space

We now study the fate of entanglement as a function of the separation between the $m$ subsystems by scaling their separations by $\lambda$, which parametrizes the evolution. At large $\lambda$, in a local physical system the state will factorise into a product form, where the description of each subsystem becomes independent, since all correlations become suppressed at large separations. If the asymptotic product state is of full rank, there exists a critical scale beyond which all entanglement vanishes. As a central application, we consider a quantum many-body system from which we extract the state of a subregion whose $m$ parts, $A = A_1 \cdots A_m$, become more separated as $\lambda$ grows, $\rho(\lambda)$. The state at infinite $\lambda$ satisfies the product form, Eq. (1), where $\rho_j$ is the reduced density matrix of subsystem

$j$ obtained by tracing out the complementary degrees of freedom. Let us take the entire system to be in equilibrium at a temperature $T \geqslant 0$, described by the Gibbs state $\rho_{AB} = \mathcal{Z}^{-1} \exp(-H/T)$. At large separation, the state of subsystem $A_1$ is given by $\rho_1 = \mathrm{Tr}_{A_2 \ldots A_m B} \sum_n p_n |E_n\rangle\langle E_n|$, where we have traced over the environment $B$, and the $(m-1)$ subsystems $A_j$. The sum runs over all eigenstates of the Hamiltonian $H$. The thermal Boltzmann probabilities are $p_n = e^{-E_n/T}/\mathcal{Z}$. We now need to determine whether $\rho_1$ has full rank. Let us first examine the restriction of the eigenstates of the Hamiltonian to $A_1$, $\rho_1^{(n)} = \mathrm{Tr}_{A_2 \ldots A_m B} |E_n\rangle\langle E_n|$. From the point of view of $A_1$, the degrees of freedom in the complement act as a bath, thus introducing statistical randomness in $\rho_1^{(n)}$. Since we take $A_1$ to be sufficiently small compared to the complement and the interactions generic (not fine-tuned), the bath generically has sufficient resources to induce statistical fluctuations that span the entire Hilbert space of $A_1$. Such ergodicity implies that $\rho_1^{(n)}$ will be of full rank. For instance, it is readily seen to occur for the majority of eigenstates as they live near the middle of the energy spectrum, and the eigenstate thermalization hypothesis [30–32] then states that the reduced density matrix on $A_1$ is approximately thermal, with the temperature determined by the energy $E_n$. The approximately thermal density matrix, obtained by restricting the Hamiltonian to $A_1$, then has full rank. Our statement, which we call the *full-rank hypothesis* (FRH), is more general: *all* eigenstates of a generic local Hamiltonian have a full-rank reduced density matrix associated with a sufficiently small subregion. In practice, the subregion should not exceed half the system. To complement the physical argument given above, we shall give an example in a frustrated quantum magnet in the next section.

Now, the reduced density matrix of subsystem 1 is the convex sum $\rho_1 = \sum_n p_n \rho_1^{(n)}$. The FRH implies that $\rho_1$ inherits full rank from the states $\rho_1^{(n)}$. (Since the sum of a full-rank matrix and an arbitrary one also has full rank, we actually only need to know that at least one $\rho_1^{(n)}$ is of full rank.) As the same holds for the other subsystems, we conclude that the $\lambda = \infty$ product has full rank, and thus lies inside the separable continent (not on its frontier). Therefore, there exists a scale beyond which all entanglement disappears. The sudden death occurs irrespective of the number of subsystems or the precise nature of the state; it even holds at a quantum critical phase transition where quantum fluctuations proliferate to all scales. This result shows that, although local observables can possess slowly decaying algebraic correlations, entanglement decays drastically faster, to the point of having a finite range. This can be understood from the monogamy of entanglement: in a typical local Hamiltonian, the interactions favour entanglement among nearby degrees of freedom, which strongly limits the capacity to entangle with more distant sites. In analogy with charged impurities in a material, one could say that "entanglement is screened" by

the environment of $A = A_1 \cdots A_m$. Our conclusion encompasses and generalises numerous examples of bipartite entanglement sudden death at finite separation [33–39], which are very specific cases of the general multipartite phenomenon. Below we illustrate the rigorous finite range of multiparty entanglement in two models.

## III. QUANTIFYING ENTANGLEMENT

In order to illustrate the Fate of Entanglement (FoE) under various evolutions, we will evaluate powerful quantifiers of entanglement, including for genuine multiparty entanglement. First, we compute a measure of 2-party entanglement, focusing on a pair of adjacent spins, called the logarithmic negativity $\mathcal{E}$ [40, 41]. In the case of two spins, $\mathcal{E} = 0$ implies that the reduced density matrix is separable [42], whereas for entangled states we have $\mathcal{E} > 0$. Second, we study multiparty entanglement via the geometric entanglement $\mathcal{D}$ [6] defined as the smallest Hilbert-Schmidt distance between the state and the separable set,

$$\mathcal{D} = \min_{\sigma \in \text{SEP}} d(\rho, \sigma), \qquad (3)$$

where the minimization is over all $\rho_{\text{sep}}$ living in the set of separable states of $m$-parties. Similarly, one can define the geometric genuine multiparty entanglement as

$$\mathcal{D}_b = \min_{\sigma \in \text{bSEP}} d(\rho, \sigma), \qquad (4)$$

where SEP and bSEP refer to the separable and biseparable sets, respectively. These are powerful quantities: $\mathcal{D}$ detect all forms of entanglement, while $\mathcal{D}_b$ detects all genuine multiparty entanglement. However, they can be difficult to evaluate due to the optimisation over a large parameter space. Nevertheless, an efficient method (due to Gilbert) exploiting the convexity of SEP and bSEP was shown to produce strong results. We will also employ a more direct global optimisation procedure, which when used in a "layered" fashion can produce better upper bounds on $\mathcal{D}, \mathcal{D}_b$ in certain cases. The values we give below are the best of the two methods for the given state, and satisfy convergence criteria (see Appendix). Finally, we also employ a much simpler criterion $W$ which indicates the presence of genuine 3-spin entanglement by detecting a property that cannot hold for biseparable states (2) [43]. It is defined from the matrix elements $\rho_{ij}$, $i, j = 1, \ldots, 8$, of a 3-spin density matrix [43]:

$$W = |\rho_{23}| + |\rho_{25}| + |\rho_{35}| - \sqrt{\rho_{11}\rho_{44}} - \sqrt{\rho_{11}\rho_{66}} - \sqrt{\rho_{11}\rho_{77}} - \frac{1}{2}(\rho_{22} + \rho_{33} + \rho_{55}). \qquad (5)$$

$W > 0$ indicates the presence of genuine 3-party entanglement, whereas for $W \leqslant 0$ a definitive conclusion cannot be made. We thus set $W = 0$ in this case. As the above expression for the criterion is basis-dependent, we maximize its value over all possible local unitary transformations of the state, $(U_1 \otimes U_2 \otimes U_3)\rho(U_1^\dagger \otimes U_2^\dagger \otimes U_3^\dagger)$, where $U_j$ is a generic $2 \times 2$ unitary matrix for the spin $j$. While it is not a proper entanglement measure, large values of $W$ generically correspond to stronger genuine 3-party entanglement. For instance, $W$ is maximal for Werner states, which are maximally-entangled 3-qubit states, and is minimal for the maximally mixed state (the identity).

In the cases considered below, we obtain values of $\mathcal{D}_{(b)}$ on the order of $10^{-6}$ beyond a certain point in the evolution, which implies that the corresponding type of entanglement is so small that it is of no practical relevance. One could stop there and move on. However, we want to make rigorous statements regarding the sudden death in order to demonstrate the general results in the strongest terms. We will thus employ an approach that can rigorously certify whether a density matrix is (bi)separable: the general trace criterion (GTC) [44]. The idea is to use filtering and subtractions to bring the original state into an ellipsoid of (bi)separable states around a reference state; the ellipsoid is strictly larger than the ball referred to above. The numerical optimisation used is very similar to the one for $\mathcal{D}_{(b)}$.

## IV. ICOSAHEDRAL MOLECULE

We first illustrate the above results with a simple quantum system: the anti-ferromagnetic Ising model on the 12-spin icosahedron, see Fig. 2a, with Hamiltonian

$$H = J \sum_{\text{bonds } \langle i,j \rangle} \sigma_i^x \sigma_j^x - h \sum_{\text{sites } i} \sigma_i^z. \tag{6}$$

The first term corresponds to an anti-ferromagnetic interaction $J > 0$ that tends to anti-align neighboring spins along $x$, while the second one is a transverse field that polarises all spins along $z$. Such a molecular quantum magnet is a combination of identical triangular faces, and thus possesses strong geometric frustration. In what follows we shall measure energy in units of the exchange coupling by setting $J = 1$. We investigate the fate of 2-, 3- and 4-spin entanglement as a function of magnetic field, temperature, time and separation

### A. Full-rank hypothesis and zero temperature phase diagram

As a first step, we checked that the FRH holds for subregions of adjacent spins with 1, 2 and 3 sites by obtaining the $2^{12}$ eigenstates via exact numerical diagonalization, see Appendix B. We

then considered the fate of entanglement with varying magnetic field at zero temperature, so that the evolution is parameterized by $s = h$, see Fig. 2b. At $h = 0$, the 2-spin reduced density matrix is a full-rank separable state, which suggests the existence of a sudden-death value $h_2^* > 0$ of the field below which the state is separable, and hence $\mathcal{E}(h < h_2^*) = 0$. We find $h_2^* \approx 0.6$. In contrast, the 3-spin reduced density matrix is not separable at small $h$ and we have $\mathcal{D} > 0$ as shown in Fig. 2b. Turning to the GME between the spins on a triangular face, we find that it becomes very small for $h < 0.65$, on the order of $10^{-6}$. Using the GTC, we are able to certify that the state is biseparable for $h \leqslant 0.25$. Although the certification becomes more difficult for larger field values, we believe that the true sudden death occurs near 0.65.

In the opposite limit of $h \to \infty$, the system becomes fully polarized and is in a pure product state. Any $n$-spin density matrix is thus a pure product state, i.e., of rank one, and lies on the boundary of the separable continent. Entanglement for a general $n$-spin density matrix is therefore expected to smoothly decrease with increasing $h$ (at $h \gg 1$) without a sudden death. These features are clearly visible in Fig. 2b for the case of 2- and 3-spin subregions. In the following we focus on the representative case $h = 3$, where 2-party and genuine 3-party entanglement are present at zero temperature. For additional data regarding the phase diagram of the model, see Appendix C.

## B. Temperature

Let us now investigate the FoE with temperature at $h = 3$. From the criterion of the separable ball around the infinite-temperature state [13], we determine the temperatures where $m$-party entanglement is guaranteed to be absent for $m = 2, 3$. We find $T_2 = 8$ and $T_3 = 22$, respectively. However, these do not tell us if entanglement is absent at lower temperatures. The logarithmic negativity $\mathcal{E}$, as shown in Fig. 2, vanishes at $T_{\mathcal{E}} = 1.792$, implying a sudden death of 2-spin entanglement at a much smaller temperature than $T_2$. Going beyond 2 spins, with the GTC we can rigorously certify that $T \geqslant T_3^* = 2.5$ leads to full separability of the 3-spin reduced density matrix. This value is close to the point where $\mathcal{D}$ becomes small, see Fig. 2c. GME is expected to disappear before generic multipartite entanglement during the evolution, and indeed we find $T_W < T_3^*$. However, we stress that the condition $W = 0$ is not informative, so GME could occur for $T$ up to $T_3^*$, but certainly not beyond. Going beyond $W$, we have used the generalisation of the GTC to certify the absence of GME beyond a certain temperature, and have obtained an exact upper bound for the death of GME: $T_{\text{GME}}^* \leqslant 2.0$. In Fig. 2c, the computed values of $\mathcal{D}_b$ show a rapid decline of GME, and provide strong evidence that our upper bound is almost tight. To

summarize, we have verified that the multipartite 3-spin entanglement, genuine or not, entirely disappears beyond some temperature, in agreement with the general expectation.

## C.   Quench

Next, we consider the evolution of entanglement with time during a quantum quench. We initialize the system in a pure product state of up and down spins in the $\sigma^z$-basis at $t = 0$ (in the following, we refer to this quench protocol as Ico-1), and let it evolve under the Ising Hamiltonian discussed above with $h = 3$, see Appendix D for details regarding the initial state. In Fig. 2d we show the evolution of various measures, including $\mathcal{D}$ for 4-spin subregion composed of two adjacent triangular faces. All quantities rapidly rise, reach a maximum, and fall to zero, with $\mathcal{D}$ for 4 spins being the longest lived. We thus see a clear illustration of the dynamical sudden death of entanglement for the case of $m = 2, 3, 4$ spins. For 3 spins, we can rigorously and tightly bound the sudden death to occur when $t_3^* \leqslant 0.369$, while for 4 spins we find $t_4^* \leqslant 1.5$. To reach a stronger conclusion regarding the fate of GME for 3 spins, we use $\mathcal{D}_b$ to obtain a rigorous upper bound on the sudden death time of GME: $t_{\mathrm{GME}}^* \leqslant 0.25$. Finally, an interesting observation is that the non-equilibrium evolution can generate stronger 2- and 3-party entanglement compared to what is found in equilibrium, even at zero temperature, see Fig. 2c. In order to show that the above behavior is generic, we have studied two other quenches for the icosahedral molecule. Ico-2 corresponds to a time evolution using the $h = 3$ Hamiltonian with a distinct initial product state of up and down spins in the $\sigma^z$ basis. Ico-3 corresponds to the sudden change of the transverse field, $h = 0.04 \rightarrow 3$; as such, the initial state possesses entanglement (see Fig.2b) in contrast to Ico-1 and Ico-2. For both Ico-2 and Ico-3 protocols, we rigorously show that the 3- and 4-spin states becomes fully separable in finite time, as for Ico-1, see Appendix D.

## D.   Separation

As a final example, we consider the evolution of entanglement with separation, working in the groundstate at $h = 3$. First, for $m = 2$ spins, we find that entanglement disappears when the spins become separated by 2 bonds. Second, for $m = 3$ spins, we scale the minimal triangle by $\lambda = 2$ so that the sides have bond-length two. We find that the state is fully separable using the GTC. We thus see a striking illustration of the short-ranged nature of entanglement, genuine or not, even in a small many-body system.

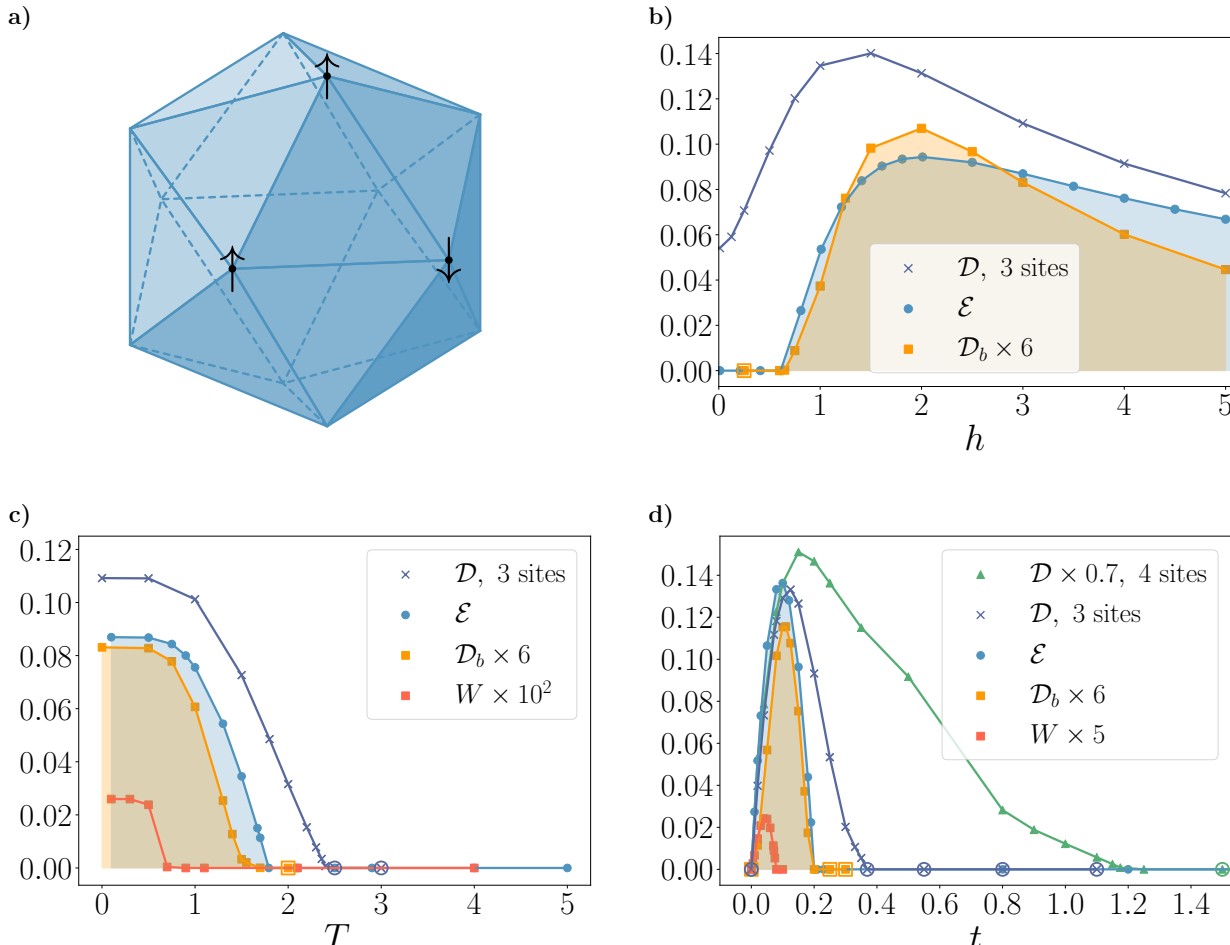

FIG. 2. **Icosahedral molecule and its entanglement evolutions. a)** Illustration of the quantum molecular magnet with icosahedral geometry. **b)** Dependence of the geometric entanglement $\mathcal{D}$, the 2-spin logarithmic negativity $\mathcal{E}$ and the genuine 3-party entanglement criterion $\mathcal{D}_b$ in the icosahedral molecule as a function of the transverse field at zero temperature. **c)** Evolution of the same quantities and the criterion $W$ as a function of temperature with $h = 3$. **d)** Same quantities as a function of time in the quench protocol Ico-1, where the system is prepared in a product state and evolves under the anti-ferromagnetic Ising Hamiltonian with $h = 3$. In panels **b)-c)-d)**, the open shapes indicate that the corresponding distance measure is rigorously certified to vanish: the state is fully (bi)separable.

## V.   SPIN CHAIN

We next consider a one-dimensional spin chain with anisotropic nearest-neighbour interactions

$$H_{\text{xyz}} = \sum_{i=1}^{N} \left( J_x \sigma_i^x \sigma_{i+1}^x + J_y \sigma_i^y \sigma_{i+1}^y + J_z \sigma_i^z \sigma_{i+1}^z - \vec{h} \cdot \vec{\sigma}_i \right) \tag{7}$$

with periodic boundary conditions such that $N + 1 \equiv N$. For our example, we randomly pick in the range $[-1, 1]$ the 3 exchange couplings $J_i$, and 3 components of the Zeemann magnetic field

$h_i$: $\vec{J} = (-0.443, 0.0938, 0.915)$ and $\vec{h} = (0.930, -0.685, 0.941)$. We work with $N = 14$ sites. The groundstate is unique with energy gap $\Delta E = 2.0007$. We have verified that the FRH holds for all $2^{14}$ eigenstates by computing the smallest eigenvalue of the reduced density matrix for a subregion made of 4 adjacent spins, as shown in Appendix B, which confirms the generic nature of $H_{\text{xyz}}$.

### A. Separation

We first examine the FoE with respect to separation in the groundstate. Two spins are separable if they are next nearest neighbours or further apart. For three spins, the GTC certifies the reduced density matrix of sites $(1, 3, 5)$ to be biseparable, while it certifies full separability for sites $(1, 3, 6)$. These findings illustrate the short-ranged nature of entanglement in equilibrium.

### B. Quench

We consider a quench by initializing the chain in a random full product state. Fig. 3 shows the temporal evolution of $\mathcal{D}, \mathcal{D}_b$ for 3 adjacent sites. Interestingly both quantities oscillate at early times, with $\mathcal{D}$ showing a small revival around $t = 5$ before finally becoming zero. Using the GTC, we can certify that for $t \geqslant 5.4$ the subregion becomes fully separable. The GME shows a faster decline, and we can certify biseparability for $t \geqslant 2.4$. We have also been able to show that the subregion of 4 adjacent spins becomes fully separable for $t \geqslant 13$ via the GTC. As above, computing $\mathcal{D}$ is easier than exact certification, and we have obtained $\mathcal{D}(t = 9) = 3.3 \times 10^{-7}$, which strongly suggests that exact separability occurs at earlier times. Finally, to put the non-equilibrium results in context, we give the geometric entanglement of 3 adjacent spins in the groundstate: $\mathcal{D} = 0.196$, which is much less than the maximal value achieved during the quench.

It would be desirable to study quenches in larger systems, where recent tensor network methods designed to track the reduced density matrix of small subregions [45] could, at least partially, allow one to study the fate of entanglement in a wider class of physical systems.

## VI. A CONTINENT WITHOUT GENUINE MULTIPARTITE ENTANGLEMENT FOR FERMIONS

The above discussion holds for systems of quantum spins. Crucially, spin operators at different sites are independent (they commute with each other) so that they obey bosonic statistics. However, there exist other particles in nature that do not commute, instead they acquire a minus sign upon

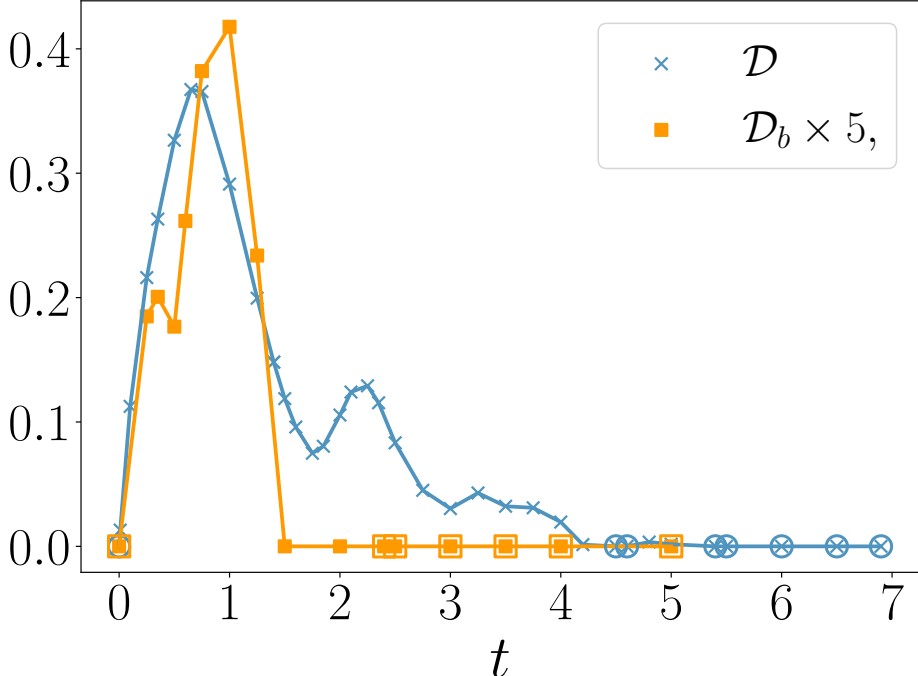

FIG. 3. **Quench for the spin chain.** Geometric entanglement $\mathcal{D}$ and $\mathcal{D}_b$ as a function of time after a quench in the spin chain defined in Eq. (7). The open shapes indicate that the state is rigorously certified to be fully (bi)separable.

exchange: fermions, like electrons or protons. Individual fermion operators possess this relative non-locality with other fermions, but observers nevertheless witness a local world since physical operators are made of an even number of fermions, and are thus bosonic. We shall consider fermions hopping on a lattice, as would pertain, for instance, to the fermionic Hubbard model that is used to describe various quantum materials. The density matrix of a physical fermionic system made of $m$ subsystems thus possesses even fermion parity. But how does fermion parity affects the definition of entanglement by constraining viable separable states? A natural notion follows from declaring that a fermionic $m$-party state is un-entangled if it can be written as $\rho_{\text{sep}}^F = \sum_k p_k \rho_1^{(k)} \otimes \rho_2^{(k)} \otimes \cdots \otimes \rho_m^{(k)}$, where each $\rho_j^{(k)}$ has even parity and the $p_k$ form a probability distribution [46–48]. This definition guarantees that the state has no quantum correlations among the $m$ components, and can thus be prepared locally. Certain states can be brought to this form, but where some $\rho_j^{(k)}$ do not have even parity. These states cannot be prepared locally and their non-local correlations can be exploited for quantum tasks such as quantum teleportation or quantum data hiding [49–51]. Moreover, we define biseparable fermionic states as states of the form (2) where each $\rho_{\mathcal{I}_k}$ and $\rho_{\overline{\mathcal{I}}_k}$ commute with the local fermionic parity of the subsystems they pertain to. Fermionic states thus possess GME

if they are not fermionic biseparable.

Given an un-entangled fermionic physical state $\rho_{\text{sep}}^F$, can we find an entanglement-free region around it as was the case for bosons? We show that the answer is no, and then analyze the same question but for the more interesting case of GME. The argument, adapted from Ref. [52] and generalized to the multipartite case, is the following. Without loss of generality, it suffices to consider the case with $m = 2$ components (we can always bipartition the initial multiparty Hilbert space) by performing a small deformation, $\rho_{\text{sep}}^F + \delta\rho$, where $\delta\rho$ has zero trace and contains terms of odd fermion parity for subsystem 1. For example, one can hop an odd number of fermions between subsystems 1 and 2, e.g., $\delta\rho = \epsilon(|01\rangle\langle10| + |10\rangle\langle01|)$ with positive $\epsilon \ll 1$. The deformed state does not have even parity for subsystem 1, and hence is not separable, even for arbitrarily small $\epsilon$. Therefore, no entanglement-free region exists around a fermionic separable state since nearby states arbitrarily close to $\rho_{\text{sep}}^F$ contain entanglement between any two subsystems. In the fermionic world, there is thus no separable continent. Pictorially, separable states lie on a zero-width sand beach, surrounded by an ocean of entangled states arbitrarily close by. However, we find that entangled states in the vicinity of fermionic separable states are fermionic biseparable by direct construction, see Appendix E. The proof involves taking an un-entangled fermionic physical state, and showing explicitly that a generic small perturbation around it can always be brought into a fermion biseparable form. This implies the existence of a fermionic biseparable continent surrounding the separable beach, and in turn that fermionic systems experience a sudden death of GME during typical evolutions with temperature, time or space. Hence, only non-GME and bipartite entanglement are robust in fermionic systems. We give a schematic illustration of these features in Fig 4.

Turning the tables around, non-GME fermionic entanglement is more robust than in bosonic systems since the state cannot land on a continent upon evolution. Let us say that we heat a fermion system that contains multipartite entanglement at low $T$. The infinite temperature state is un-entangled, but non-GME entanglement will decay gradually as a function of temperature without a sudden death. This behavior was observed for integer quantum Hall states [53] and free-fermion systems [46, 54]. Analogous conclusions hold for temporal or spatial evolution. For instance, the bipartite entanglement between disjoint regions in fermionic quantum critical systems in arbitrary dimensions decays as a power-law with the distance, without a sudden death [39]. In contrast, we predict that any measure of GME in such systems will suffer a sudden death.

The geometry of the space of states acquires a distinct structure due to the presence of a superselection rule, which is due to the fermion parity symmetry. By the same argument, a similar

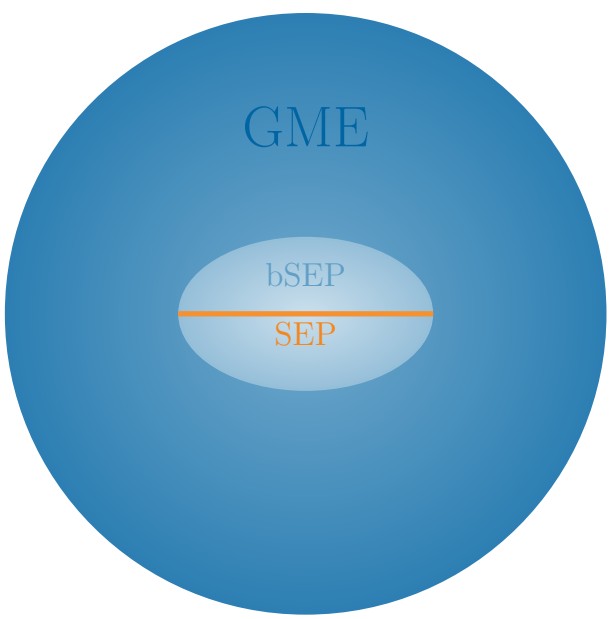

FIG. 4. **Structure of the space of states for fermions.** For fermions, because of the fermion-parity superselection rule, there is no separable continent. Instead, separable states form a zero-width sand beach with entangled states arbitrarily close. This beach is surrounded by a light-blue region of biseparable fermionic states, beyond which states possess GME.

structure arises with other superselection rules, where a symmetry becomes enforced. Requiring the symmetry to hold for subsystems constrains states that can appear in separable decompositions, thus leading to the disappearance of separable continents. Symmetry-enforced entanglement is thus more resilient.

## VII. OUTLOOK

In this paper, we discussed how the structure of the space of physical states, and in particular the presence of a separable continent, leads to the sharp disappearance of entanglement in various physical situations, under very generic assumptions. For fermionic systems, the parity superselection rule forbids the existence of a separable continent and modifies the fate of entanglement. Interestingly, we showed that there is a continent without genuine multipartite fermionic entanglement, which is the richest form of entanglement. So, although fermionic systems do not experience an entanglement sudden death, their genuine multipartite entanglement suffers the same fate as for bosons: it irreversibly disappears during evolution. The same conclusion holds for other types of superselection rules, providing a generic recipe for the creation of robust entangled states.

These new insights regarding the structure of entanglement in many-body states will have strong impact for quantum simulation and computation. For instance, efficient algorithms used to simulate quantum matter should incorporate the fact that distant particles are not entangled in spin/boson systems, which strongly constrains the variational space of many-body wave functions. Our work also paves the way for numerous outstanding research avenues. For instance, it will be essential to investigate how multipartite entanglement evolves in a plethora of realistic model Hamiltonians, and to determine the criteria that govern its demise under evolution. It would also be desirable to investigate systems which could escape the fate of entanglement. In parallel, we need a better understanding of violations of the FRH by certain systems with topological order or quantum many-body scars, for instance. Finally, we have also seen examples of how quantum matter out-of-equilibrium generates stronger multipartite entanglement than in equilibrium. This is the tip of iceberg: we expect that non-equilibrium dynamics can lead to rich entanglement structures that await to be discovered.

### Acknowledgements

We are grateful to Liuke Lyu for discussions and sharing code. G.P. holds an FRQNT Postdoctoral Fellowship, and acknowledges support from the Mathematical Physics Laboratory of the Centre de Recherches Mathématiques. W.W.-K. is supported by a grant from the Fondation Courtois, a Chair of the Institut Courtois, a Discovery Grant from NSERC, and a Canada Research Chair.

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

## Appendix A: Entanglement measures and criteria

We give the definitions of the entanglement-related quantities we discuss in the main text, namely the logarithmic negativity $\mathcal{E}$, the geometric entanglement $\mathcal{D}$ and the genuine 3-party entanglement criterion $W$.

### 1. Logarithmic negativity

We consider the density matrix $\rho$ matrix pertaining to two subsystems 1 and 2. The logarithmic negativity [40, 41] is

$$\mathcal{E} = \log \left\| \rho^{T_1} \right\| \tag{A1}$$

where $\|X\| = \mathrm{Tr}\sqrt{XX^\dagger}$ is the trace-norm, and $\rho^{T_1}$ is the partially-transposed density matrix with respect to subsystem 1. This entanglement measure is related to the Peres separability criterion for density matrices [55]. In full generality, a vanishing logarithmic negativity is only a necessary condition for separability, but in the case of two spins (or qubits), it is both necessary and sufficient [42].

### 2. Geometric entanglement

The geometric entanglement [6] is the distance between the state $\rho$ and the closest separable state,

$$\mathcal{D} = \min_{\rho_{\mathrm{sep}}} \sqrt{\mathrm{Tr}(\rho - \rho_{\mathrm{sep}})^2}. \tag{A2}$$

$\mathcal{D}_b$ is analogously defined as the distance to the biseparable set. For a small number of spins, we can perform the optimization to get good upper bounds. We have used $\mathcal{D}_{(b)}$ for 3 spins, while the Gilbert algorithm was used to obtain $\mathcal{D}$ for 4 spins owing to the larger parameter space. In performing the optimization for $\mathcal{D}$ in the case of 3 spins, we search over the set $\sum_{i=1}^{u} K_i$ with a fixed number of components $u$, where $K_i = \rho_1^i \otimes \rho_2^i \otimes \rho_3^i$ is a general product state (not necessarily pure). We begin with a low $u$, such as $u_1 = 5$, find the optimum by a brute force search (we employed the global routine 'fmincon' in Matlab) using many initial points. Once the closest separable state is found, we used as the starting guess for the search with a larger number of components, $u_2 = u_1+1$. Once the variation of $\mathcal{D}$ between successive $u$-values becomes less than a given threshold, we stop the process. We have found that we can get excellent upper bounds by going up to $u = 12$. The

values obtained were cross-checked with the Gilbert approach. An analogous approach yields $\mathcal{D}_b$, but now the product states are replaced by $\rho_1 \otimes \rho_{23} + \sigma_2 \otimes \sigma_{13} + \varrho_3 \otimes \varrho_{12}$, where the subscript denotes the spins of the corresponding physical density matrix.

The separability/biseparability certification through the GTC is obtained via an optimization over the same $u$-component set, as described in [44].

## 3. Genuine 3-party entanglement criterion

The criterion $W$ is defined from the matrix elements $\rho_{ij}$, $i, j = 1, \ldots, 8$, of a 3-spin density matrix. It reads [43]

$$W = |\rho_{23}| + |\rho_{25}| + |\rho_{35}| - \sqrt{\rho_{11}\rho_{44}} - \sqrt{\rho_{11}\rho_{66}} - \sqrt{\rho_{11}\rho_{77}} - \frac{1}{2}(\rho_{22} + \rho_{33} + \rho_{55}). \qquad \text{(A3)}$$

Positive $W > 0$ indicates the presence of genuine 3-party entanglement, whereas for $W \leqslant 0$ a definitive conclusion cannot be made. We thus define $W = 0$ in this case. The criterion $W$ is basis-dependent. Numerically, we thus maximize its value over all possible local unitary transformations $(U_1 \otimes U_2 \otimes U_3)\rho(U_1^\dagger \otimes U_2^\dagger \otimes U_3^\dagger)$, where $U_j$ is a generic $2 \times 2$ unitary matrix for the spin $j$. While it is not an entanglement quantifier, large values of $W$ generically correspond to stronger genuine 3-party entanglement. For instance, $W$ is maximal for Werner states, which are maximally-entangled 3-qubit states, and is minimal for the maximally mixed state (the identity).

## Appendix B: Test of the full-rank hypothesis

We test the full-rank hypothesis (FRH) for the icosahedral molecule at a representative value of the transverse field, $h = 3$, for three adjacent sites, as well as for four adjacent sites in the spin chain. In Fig. 5 we plot the minimal eigenvalues $\lambda_{\min}$ of the 3- and 4-spin reduced density matrices for each eigenstate of the Hamiltonian, labeled by the corresponding energy $E$. All the minimal eigenvalues are strictly positive, which indicates that the FRH is satisfied. For the icosahedral molecule, the groundstate has the smallest $\lambda_{\min}$, whereas it is the second smallest in the spin chain.

## Appendix C: Phase diagram of the molecular quantum magnet

We report the zero-temperature connected correlation functions of adjacent spins in the anti-ferromagnetic Ising model on the icosahedral molecule as a function of the transverse field $h$, see

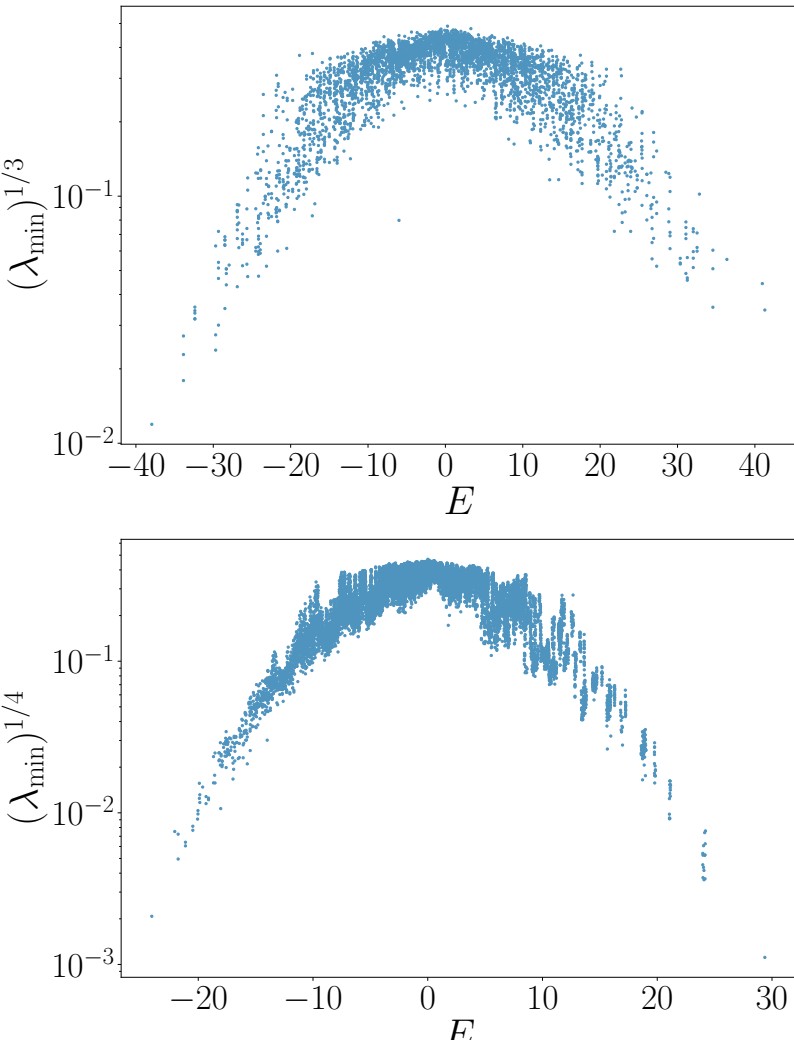

FIG. 5. **Test of the full-rank hypothesis.** Minimal eigenvalue $\lambda_{\min}$ of the reduced density matrices for three adjacent spins in the icosahedral molecule at $h = 3$ (top), and for four adjacent spins in the spin chain (bottom) for the whole spectrum, labeled by the corresponding energy $E$. We note that the groundstate has the smallest $\lambda_{\min}$ in the top panel, while it is the second smallest in the bottom one.

Fig. 6. By symmetry we have $\langle \sigma^{x,y} \rangle = 0$. The point $h = 3$ corresponds to a generic point of the phase diagram.

## Appendix D: Quench protocols

In the quench protocols Ico-1 and Ico-2, we initialize the system in a pure product state of up and down spins in the $\sigma^z$ basis, and let it evolve under the icosahedron Ising Hamiltonian discussed above with $h = 3$. We illustrate the initial states in Fig. 7. This is a planar representation of the

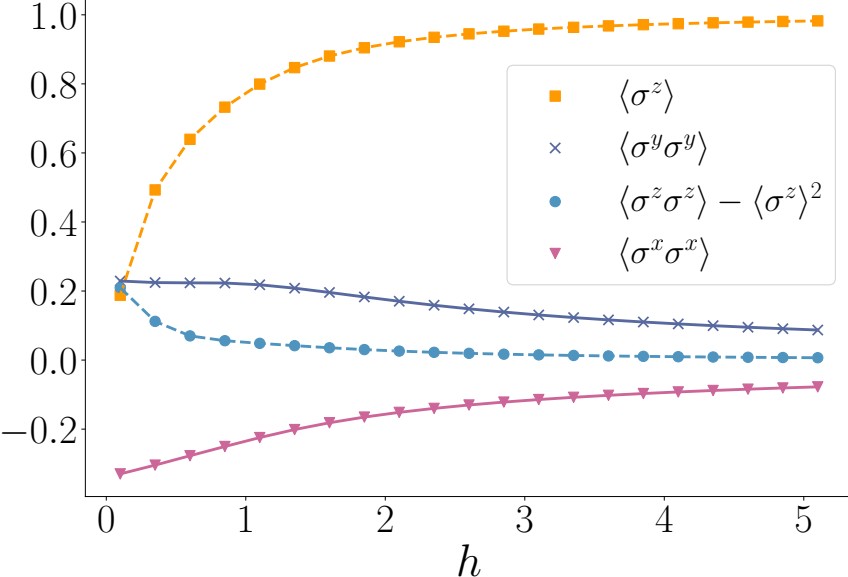

FIG. 6. **Spin correlations in the icosahedral molecule.** Various spin correlation functions versus the transverse field in the icosahedral molecule at zero temperature.

icosahedral molecule, where blue sites are initialized in a spin-up state, whereas orange one are initialized in a spin-down state. The colored face represents the 3-spin subsystem for which we compute various metrics during the time evolution. For the Ico-3 protocol, the initial state is the groundstate of the Ising Hamiltonian with $h = 0.04$ and it evolves under the Hamiltonian with $h = 3$. The time-evolution of the geometric entanglement $\mathcal{D}$ for both Ico-2 and Ico-3 quenches is represented in Fig. 8. We have further studied the 4-spin subregion made of two adjacent triangular faces, and were able to certify full separability for $t \geqslant 4$ (Ico-2) and $t \geqslant 1.5$ (Ico-3).

## Appendix E: Genuine multipartite entanglement for fermions

In this appendix, we show that fermionic separable states are surrounded in their close vicinity by regions of fermionic biseparability, implying the sudden death of genuine multipartite entanglement (GME) in fermionic systems.

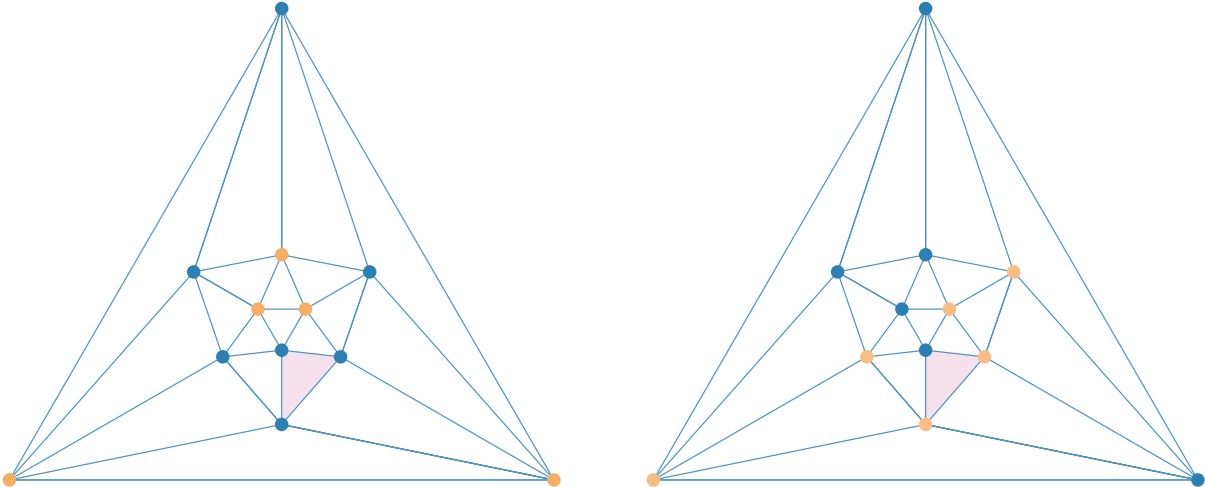

FIG. 7. **Initial states in the quench protocols.** Blue sites are initialized in a spin-up state, whereas orange ones are initialized in a spin-down state. We study the colored face during the time evolution. The left panel represents the initial stat for the Ico-1 protocol, and the left panel pertains to the Ico-2 protocol.

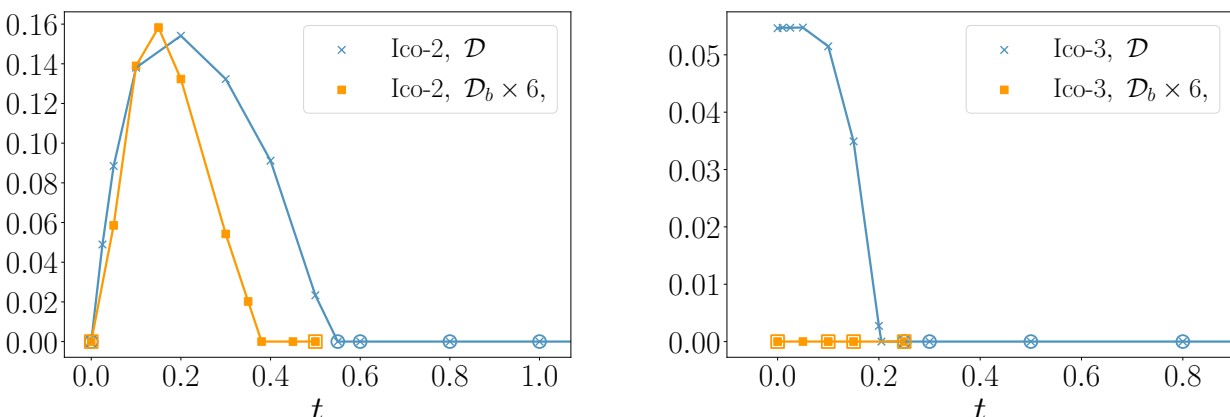

FIG. 8. **Icosahedron quench protocols Ico-2 and Ico-3.** The geometric entanglement $\mathcal{D}$ and $\mathcal{D}_b$ for the quench protocols Ico-2 and Ico-3. The open shapes indicate that the 3-spin state is rigorously certified to be fully (bi)separable.

### 1. Three parties

Let us first consider the case of $m = 3$ fermionic modes. A fermionic separable state has the form

$$\rho_{\text{sep}}^F = \sum_k p_k \, \rho_1^{(k)} \otimes \rho_2^{(k)} \otimes \rho_3^{(k)} \tag{E1}$$

with $p_k \geqslant 0$ and $\sum_k p_k = 1$. The 1-mode states $\rho_j^{(k)}$ are normalized, Hermitian and positive semi-definite operators with even local fermion parity: $(-1)^{n_j} \rho_j^{(k)} (-1)^{n_j} = \rho_j^{(k)}$, where the number

operator $n_j = c_j^\dagger c_j$ is defined in terms of the mode annihilation operator $c_j$. The generic form for such states is

$$\rho_j^{(k)} = \frac{1}{2}\mathbb{I} + a_j^{(k)}\sigma^z, \qquad -\frac{1}{2} \leqslant a_j^{(k)} \leqslant \frac{1}{2}. \tag{E2}$$

We take $\rho_{\text{sep}}^F$ to lie in the interior of the fermion separable set, for otherwise entangled states would lie in the immediate vicinity of the state. By inclusion, $\rho_{\text{sep}}^F$ also lies in the interior of the bosonic separable continent. Using convexity of separable states, we are thus guaranteed that there exists a region surrounding $\rho_{\text{sep}}^F$ where states have a (bosonic) separable form. We parametrize them by

$$\rho(\epsilon) = \sum_k p_k \ (\rho_1^{(k)} + \epsilon\, \omega_1^{(k)}) \otimes (\rho_2^{(k)} + \epsilon\, \omega_2^{(k)}) \otimes (\rho_3^{(k)} + \epsilon\, \omega_3^{(k)}) \tag{E3a}$$

with

$$\omega_j^{(k)} = b_j^{(k)}\sigma^x + d_j^{(k)}\sigma^y, \tag{E3b}$$

where $\epsilon \geqslant 0$ is a positive but small parameter. These states satisfy $\rho(0) = \rho_{\text{sep}}^F$. In principle, one could consider a more generic perturbation where $\omega_j^{(k)}$ also contains a $\sigma^z$ component. However, this would be equivalent to considering a state $\tilde{\rho}(\epsilon)$ as in (E3a) but centered around another fermionic separable state $\tilde{\rho}_{\text{sep}}^F$. We thus consider the case where $\omega_j^{(k)}$ only contains terms breaking the local fermion parity. We stress that the 1-mode states in (E3a) break local fermion parity, and therefore $\rho(\epsilon)$ corresponds to an entangled fermionic state.

The constants $b_j^{(k)}$ and $d_j^{(k)}$ in (E3b) are not entirely free however. They must be chosen such that (i) the individual states in the convex combination (E3a) are positive semi-definite (they are normalized and Hermitian by definition), and (ii) the total state commutes with the total fermion parity, $(-1)^n$, where $n = \sum_j n_j$. These conditions translate to, respectively,

$$(a_j^{(k)})^2 + \epsilon^2(b_j^{(k)})^2 + \epsilon^2(d_j^{(k)})^2 \leqslant \frac{1}{4} \tag{E4a}$$

and

$$\sum_k p_k\epsilon\left(\omega_1^{(k)} \otimes \rho_2^{(k)} \otimes \rho_3^{(k)} + \rho_1^{(k)} \otimes \omega_2^{(k)} \otimes \rho_3^{(k)} + \rho_1^{(k)} \otimes \rho_2^{(k)} \otimes \omega_3^{(k)}\right)$$
$$+ \sum_k p_k\epsilon^3\omega_1^{(k)} \otimes \omega_2^{(k)} \otimes \omega_3^{(k)} = 0\,. \tag{E4b}$$

The first condition (E4a) can always be satisfied for small enough $\epsilon$, unless the local fermionic state in $\rho_{\text{sep}}^F$ is $\rho_j^{(k)} = \frac{1}{2}(\mathbb{I} \pm \sigma^z)$. In those situations, no parity-breaking perturbation on $\rho_j^{(k)}$ is allowed and $\omega_j^{(k)} = 0$.

Because all the odd-parity terms have to cancel, see (E4b), we recast $\rho(\epsilon)$ as

$$\rho(\epsilon) = \rho_{\text{sep}}^F + \epsilon^2 \sum_k p_k \left( \rho_1^{(k)} \otimes \omega_2^{(k)} \otimes \omega_3^{(k)} + \omega_1^{(k)} \otimes \rho_2^{(k)} \otimes \omega_3^{(k)} + \omega_1^{(k)} \otimes \omega_2^{(k)} \otimes \rho_3^{(k)} \right). \qquad (E5)$$

For each $k$ in the sum for $\rho(\epsilon)$, we recast the corresponding term (dropping the indices and tensor product symbols for readability) as

$$\rho\rho\rho + \epsilon^2(\rho\omega\omega + \omega\rho\omega + \omega\omega\rho) = \frac{1}{3}\rho(\rho\rho + 3\epsilon^2\omega\omega) + \frac{1}{3}(\rho\rho\rho + 3\epsilon^2\omega\rho\omega) + \frac{1}{3}(\rho\rho + 3\epsilon^2\omega\omega)\rho. \qquad (E6)$$

We thus have

$$\rho(\epsilon) = \sum_k \frac{p_k}{3} \left( \rho_1^{(k)} \otimes \rho_{23}^{(k)}(\epsilon) + \rho_{13}^{(k)}(\epsilon) \otimes \rho_2^{(k)} + \rho_{12}^{(k)}(\epsilon) \otimes \rho_3^{(k)} \right) \qquad (E7a)$$

with

$$\rho_{ij}^{(k)}(\epsilon) = \rho_i^{(k)} \otimes \rho_j^{(k)} + 3\epsilon^2 \omega_i^{(k)} \otimes \omega_j^{(k)}. \qquad (E7b)$$

For $0 \leqslant \epsilon \leqslant \epsilon^*$ and $\epsilon^* > 0$ small enough, this state is guaranteed to be positive semi-definite for all $i, j, k$. The only potential obstacle would be if one $\rho_j^{(k)}$ were of the form $\frac{1}{2}(\mathbb{I} \pm \sigma^z)$ (with a vanishing eigenvalue), but in that case the corresponding perturbation would have to vanish, $\omega_j^{(k)} = 0$, as explained above. Moreover, it is direct to show that condition (E4a) is fulfilled. We thus conclude that for $0 \leqslant \epsilon \leqslant \epsilon^*$, the state $\rho(\epsilon)$ has a biseparable form (2), where each state involved in the convex combination is normalized, Hermitian, positive semi-definite and commutes with its local fermion parity. It it thus fermionic biseparable.

## 2. Four parties

For $m = 4$, the argument is similar. The even terms in the sum for $\rho(\epsilon)$, following the notation of (E6), have the form

$$\rho\rho\rho\rho + \epsilon^4\omega\omega\omega\omega + \epsilon^2(\rho\rho\omega\omega + \text{perm.}) =$$
$$\frac{1}{7}(\rho\rho\rho\rho + 7\epsilon^4\omega\omega\omega\omega) + \left\{ \frac{1}{7}(\rho\rho\rho\rho + 7\epsilon^2\rho\rho\omega\omega) + \text{perm.} \right\}. \qquad (E8)$$

We recast the terms as

$$\rho\rho\rho\rho + 7\epsilon^4\omega\omega\omega\omega = \frac{1}{2}(\rho\rho + \sqrt{7}\epsilon^2\omega\omega)(\rho\rho + \sqrt{7}\epsilon^2\omega\omega) + \frac{1}{2}(\rho\rho - \sqrt{7}\epsilon^2\omega\omega)(\rho\rho - \sqrt{7}\epsilon^2\omega\omega),$$
$$(E9)$$

$$\rho\rho\rho\rho + 7\epsilon^2\rho\rho\omega\omega = \rho\rho(\rho\rho + 7\epsilon^2\omega\omega),$$

and similarly for the other five permutations involving two $\rho$ and two $\omega$. As for the case $m = 3$, there are small but non-vanishing values of $\epsilon$ for which all states above are positive semi-definite. Since they commute with their local parity, it means that the total state $\rho(\epsilon)$ is fermionic biseparable. The argument readily generalises to arbitrary values of $m \geqslant 3$.