# Peer review of "The Fate of Entanglement"

_SciPost Physics_

## Round 1 · Referee Report · Anonymous (Referee 1) · 2025-7-29

Strengths

  • unifying picture of entanglement scaling in several interesting situations
  • very clear and readable manuscript

Weaknesses

  • several of the theoretical tools employed were already developed.

Report

The authors discuss the fate of multipartite entanglement in several physically interesting situations, such as finite temperature and time evolution. They also investigate the scaling of multipartite entanglement as a function of the spatial separation between different regions. By using known results in quantum information, the authors argue that quite generically multipartite entanglement decays and the system becomes separable. This statement is supported by exact diagonalization data for an Ising model on the icosahedral molecule, and for the Ising spin chain.

The main strength of the paper is that it suggests that most forms of collective entanglement are surprisingly fragile, disappearing under common physical processes such as heating, time evolution, and environmental decoherence. Although most of the theoretical tools that are used in the paper were already developed in the quantum info community, they are employed here in a wide variety of physically interesting situations. I think that the general overview provided in the paper could contribute to develop a unifying picture of multipartite entanglement scaling. Moreover, the paper is quite well written and the material is quite nicely presented. For this reason I recommend the paper for publication in Scipost Physics

Recommendation

Publish (easily meets expectations and criteria for this Journal; among top 50%)

---

## Round 1 · Referee Report · Anonymous (Referee 2) · 2025-10-21

Strengths

Strengths: 1. Long time/distance strict vanishing of true entanglement in a large variety of situations is an exciting result 2. Proofs are easy to follow and framed in a simple language.

Weaknesses

Weaknesses: 1. The organization of the paper is sometimes confusing. 2. Some details on the order of limits/some counterexamples could be more discussed.

Report

In this article, the authors present a unifying point of view on the fate of (multipartite) entanglement at long time/distance/large temperature. It can be summarized as follows: Given a multipartition of a (sub)system in m parts, the genuine entanglement between these parts vanishes strictly after a finite time. The results also stand at a finite temperature (when considering thermal states) or a finite distance (when considering an eigenstate).

The authors extend then their approach to fermionic systems, with a small caveat. While entanglement does not vanish at fixed time due to the conservation of the fermionic parity, genuine multipartite entanglement still does.

The authors have demonstrated clearly a surprising fragility of entanglement in physical systems. Their result open numerous questions related to decoherence and the survival of quantum information, and as they mention in their concluding remarks, may open new pathways for an efficient simulation of quantum systems. While I think most of the ideas are not technically entirely new, they are framed in a clear light and given a rigorous origin. As such I recommend the publication of the article in SciPost Physics after addressing a few comments and questions.

Comments and questions: 1. I found that the precise limits for which the results of the article are valid confusing. In particular, I think that it would be useful to clarify that the results stand for a subsystem consisting in a finite number of spins (or a finite number of such subsystems).

  1. While relatively trivial, for the icosahedron (and in generally all finite systems), should we not expect that at time far greater than the Heisenberg time we have a revival of entanglement? I agree this is not really physically relevant.

  2. A discussion, at least in the conclusions, is missing on some non-trivial possible counter-examples to the mechanisms studied in this paper. In particular: what happens in a) systems with topological order, b) many-body scars. Additionally (c), if one performs a local quench, we often expect slow hydrodynamics, with power-law decay of several observables. How do these examples fall in or out of the authors claims of universality?

  3. Coming back to the number of spins involved: the island of separability is valid only for finite m. And in general, its radius (at least from the point of view of maximal smallest eigenvector) decrease exponentially with the Hilbert space dimension. What is the expected scaling of the parameters at which m-multipartite entanglement disappear? And what is its effective scaling in realistic systems?

  4. Could you make a link with the numerical approaches suggested in Phys. Rev. Lett. 132, 100402 (2024) or SciPost Phys. 13, 080 (2022) / Phys. Rev. A 112, 022221 (2025)?

Requested changes

See above, in particular questions 1 and 3. A short estimate for 4 would also be appreciated.
Comments 2 and 5 are more for the sake of curiosity, and do not necessarily require an answer within the article.

Recommendation

Ask for minor revision

  • validity: top
  • significance: high
  • originality: high
  • clarity: good
  • formatting: good
  • grammar: good

Author:  Gilles Parez  on 2025-11-12  [id 6025]

(in reply to Report 2 on 2025-10-21)
Category:
answer to question

Dear Referee,

We thank you for your report and valuable comments, which we address below.

  • We agree that our results, as formulated, apply to subsystems of finite Hilbert space dimension, even when the total system is infinite in the thermodynamic limit. We have added a clarifying sentence at the end of the paragraph below Eq. (1) to make this explicit.

  • It is indeed true that quenches of finite closed quantum systems will show revivals. However, this will typically occur at such large times that it is not very relevant. In practice, small subregions (compared to the complement) effectively thermalize, and the loss of entanglement holds for very long periods of time unless there is some fine-tuning in the quench. For instance, in the Icosahedral Ising model, we have not seen any revivals up to times of order $10^3$ (in units of 1/J).

  • A discussion of possible counterexamples is indeed valuable. In Sec. II.C we have clarified that our general discussion pertains to global quenches, and we have added a short discussion at the end of the first paragraph of the section addressing the case of local quenches. Whether local quenches can indeed evade the fate of entanglement discussed here remains an open question for future work. Concerning topological order and many-body scars, we have added a corresponding remark in the conclusion.

  • The question of how the fate of entanglement behaves with increasing number of spins is indeed a very important one. As correctly pointed out, the island of separability shrinks for larger $m$, implying that $m$-partite entanglement becomes more robust as $m$ increases. For instance, in Fig. 2(c), the critical temperature for 3-spin separability is higher than that for 2-spin separability, and the same holds for the characteristic times in Fig. 2(d). Our approach in this paper is to establish general and robust statements for $m$-partite separability at fixed $m$, and then illustrate them with examples. We have not analyzed in detail the scaling of the critical parameters—e.g. temperature $T_m^*$ or time $t_m^*$—with $m$. While this is indeed an important question deserving future investigation, it lies beyond the present scope of our paper. Second, if we fix the number of parties $m$, but increase the number of qubits within each party $n$ by making the subregion larger, it is an important question to understand how the results scale with increasing $n$.

  • The papers you mention are interesting. In particular the methods mentioned in the PRL could lead to efficiently getting the RDM of a small subregion as a function of time, which would allow one to study its Fate of Entanglement. We have added a comment and a reference at the end of Sec. V.B.

Sincerely,

The authors

---

## Round 2 · Author Response

Dear Referee,

We thank you for your report and valuable comments, which we address below.

  • We agree that our results, as formulated, apply to subsystems of finite Hilbert space dimension, even when the total system is infinite in the thermodynamic limit. We have added a clarifying sentence at the end of the paragraph below Eq. (1) to make this explicit.

  • It is indeed true that quenches of finite closed quantum systems will show revivals. However, this will typically occur at such large times that it is not very relevant. In practice, small subregions (compared to the complement) effectively thermalize, and the loss of entanglement holds for very long periods of time unless there is some fine-tuning in the quench. For instance, in the Icosahedral Ising model, we have not seen any revivals up to times of order $10^3$ (in units of 1/J).

  • A discussion of possible counterexamples is indeed valuable. In Sec. II.C we have clarified that our general discussion pertains to global quenches, and we have added a short discussion at the end of the first paragraph of the section addressing the case of local quenches. Whether local quenches can indeed evade the fate of entanglement discussed here remains an open question for future work. Concerning topological order and many-body scars, we have added a corresponding remark in the conclusion.

  • The question of how the fate of entanglement behaves with increasing number of spins is indeed a very important one. As correctly pointed out, the island of separability shrinks for larger $m$, implying that $m$-partite entanglement becomes more robust as $m$ increases. For instance, in Fig. 2(c), the critical temperature for 3-spin separability is higher than that for 2-spin separability, and the same holds for the characteristic times in Fig. 2(d). Our approach in this paper is to establish general and robust statements for $m$-partite separability at fixed $m$, and then illustrate them with examples. We have not analyzed in detail the scaling of the critical parameters—e.g. temperature $T_m^*$ or time $t_m^*$—with $m$. While this is indeed an important question deserving future investigation, it lies beyond the present scope of our paper. Second, if we fix the number of parties $m$, but increase the number of qubits within each party $n$ by making the subregion larger, it is an important question to understand how the results scale with increasing $n$.

  • The papers you mention are interesting. In particular the methods mentioned in the PRL could lead to efficiently getting the RDM of a small subregion as a function of time, which would allow one to study its Fate of Entanglement. We have added a comment and a reference at the end of Sec. V.B.

Sincerely,

The authors

---

## Editorial Decision

accepted_in_target_journal